# Valid and Reliable Assessment of Upper Respiratory Tract Specimen Collection Skills during the COVID-19 Pandemic

**DOI:** 10.3390/diagnostics11111987

**Published:** 2021-10-26

**Authors:** Tobias Todsen, Anne Bohr, Lisette Hvid Hovgaard, Rebekka Consuelo Eið, Thomas Benfield, Morten B. S. Svendsen, Nikolai Kirkby, Lars Konge, Christian von Buchwald, Jacob Melchiors, Martin Tolsgaard

**Affiliations:** 1Department of Otorhinolaryngology, Head and Neck Surgery and Audiology, Rigshospitalet-Copenhagen University Hospital, 2100 Copenhagen, Denmark; annebohr@hotmail.com (A.B.); rebekka_n@hotmail.com (R.C.E.); christian.von.buchwald@regionh.dk (C.v.B.); jacob.melchiors@regionh.dk (J.M.); 2Copenhagen Academy for Medical Education and Simulation, Capital Region, 2100 Copenhagen, Denmark; morten.bo.soendergaard.svendsen@regionh.dk (M.B.S.S.); Lars.Konge@regionh.dk (L.K.); martintolsgaard@gmail.com (M.T.); 3Department of Clinical Medicine, University of Copenhagen, 2100 Copenhagen, Denmark; Thomas.Lars.Benfield@regionh.dk; 4Department of Otorhinolaryngology and Maxillofacial Surgery, Zealand University Hospital, 4600 Køge, Denmark; lisette@hovgard.dk; 5Department of Infectious Diseases, Copenhagen University Hospital, Amager and Hvidovre, 2650 Hvidovre, Denmark; 6Department of Clinical Microbiology, Rigshospitalet, Copenhagen University Hospital, 2100 Copenhagen, Denmark; Nikolai.Kirkby@regionh.dk; 7Department of Obstetrics, Rigshospitalet, Rigshospitalet, Copenhagen University Hospital, 2100 Copenhagen, Denmark

**Keywords:** COVID-19 testing, SARS-CoV-2 sample, upper respiratory tract specimens, skills assessment, validation

## Abstract

Proper specimen collection is the most important step to ensure accurate testing for the coronavirus disease 2019 (COVID-19) and other infectious diseases. Assessment of healthcare workers’ upper respiratory tract specimen collection skills is needed to ensure samples of high-quality clinical specimens for COVID-19 testing. This study explored the validity evidence for a theoretical MCQ-test and checklists developed for nasopharyngeal (NPS) and oropharyngeal (OPS) specimen collection skills assessment. We found good inter-item reliability (Cronbach’s alpha = 0.76) for the items of the MCQ-test and high inter-rater reliability using the checklist for the assessment of OPS and NPS skills on 0.86 and 0.87, respectively. The MCQ scores were significantly different between experts (mean 98%) and novices (mean 66%), *p* < 0.001, and a pass/fail score of 91% was established. We found a significant discrimination between checklist scores of experts (mean 95% score for OPS and 89% for NPS) and novices (mean 50% score for OPS and 36% for NPS), *p* < 0.001, and a pass/fail score was established of 76% for OPS and 61% for NPS. Further, the results also demonstrated that a group of non-healthcare educated workers can perform upper respiratory tract specimen collection comparably to experts after a short and focused simulation-based training session. This study, therefore, provides validity evidence for the use of a theoretical and practical test for upper respiratory specimens’ collection skills that can be used for competency-based training of the workers in the COVID-19 test centers.

## 1. Introduction

Testing for the novel severe acute respiratory syndrome coronavirus 2 (SARS-CoV-2) causing the coronavirus disease 2019 (COVID-19) is an essential part of the pandemic control [1]. Millions of individuals are tested daily as a part of mass testing strategies to identify and isolate asymptomatic COVID-19 cases in society [2]. For initial SARS-CoV-2 infections, an upper respiratory tract specimen is recommended to be collected for molecular or antigen testing. The World Health Organization (WHO) recommends performing oropharyngeal (OPS) and nasopharyngeal swabbing (NPS) to ensure the collection of representative upper respiratory tract specimens for COVID-19 testing [3]. Proper specimen collection is considered the most important step in the diagnostic work-up of COVID-19 and suboptimal technique may lead to false-negative test results [4]. The test procedures can be technically difficult to perform, and due to the significant and unmet need for healthcare workers at the many community-based COVID-19 test sites, non-healthcare workers may also be used to collect upper respiratory tract specimens [5,6]. Therefore, it is imperative to ensure that the test administrators have the necessary skills to collect high-quality clinical specimens, and competency-based training should provide the skills needed. A pre-requisite for competency-based training is an assessment tool with established validity evidence for evaluating upper respiratory specimen sampling techniques at the COVID-19 test sites.

This study aims to develop and examine the validity of theoretical and practical skills tests for obtaining nasopharyngeal (NPS) and oropharyngeal (OPS) specimen collection for COVID-19 testing.

## 2. Materials and Methods

We conducted an experimental study to assess skills in obtaining NPS and OPS using Messick’s validity framework, which complies with the Standards for Educational and Psychological Testing guidelines [7]. A multiple-choice test (MCQ) and checklists following the World Health Organization (WHO) recommendations to assess NPS and OPS performance were constructed, and experts and novices were invited to participate in a validation study. Psychometric data from the study were used to explore five sources of validity evidence according to Messick’s validity framework: content evidence, response process, internal structure, relationships with other variables, and test consequences.

### 2.1. Developing the Theoretical and Practical Test

A MCQ and a checklist for upper respiratory tract specimens’ collection skills (URTS) were developed as an objective structured clinical examination (OSCE) format test with content evidence obtained through two steps. First, we used the technical guide from the WHO together with an additional literature review to ensure the contents of the URTS MCQ, and the URTS checklist followed international recommendations [3,8,9,10,11,12,13,14,15,16]. We then recruited a multidisciplinary board of experts constituted by a consultant in rhinology (CvB), a consultant in head and neck surgery (JM), a consultant in infectious disease (TB), a microbiology specialist (NK), and a medical education scientist (MT) to review the content of each item on the checklist. The items were revised and re-evaluated until consensus was reached among the expert panel.

### 2.2. Participants

Participants without any formal healthcare education were recruited to participate in the study as novices representing the competence level of general staff that could be hired in public-based COVID-19 sites for mass-testing. Their performances were assessed in a simulated setting before receiving OPS and NPS training and again afterward to measure the effect of the training session. We invited consultants and registrars (PGY-2 or more) in otolaryngology to participate in the study as the expert group. These were chosen as experts due to their anatomical knowledge and experience with procedures within the upper respiratory tract.

### 2.3. Test Setup

The theoretical test was constructed as MCQ questions with three possible answers, with only one answer being correct. Some of the questions also included pictures or anatomical animations. Printed papers with the MCQ questions were provided to all the participants who were not allowed to use handbooks, access web resources, or ask for help during the MCQ test. Each correct answer gave one point, whereas incorrect answers gave zero points.

The practical test with OPS and NPS sampling techniques was conducted using a simulation-based context. A checklist was developed for the OPS and NPS sample assessing the different sub-elements of the procedures performed correctly (one point) or incorrectly (zero points). The oropharyngeal sampling was performed on a life-sized airway demonstration model (Airsim Advance Crico, Trucorp, Belfast, Northern Ireland) with a flexible endoscope (aScope 4 RhinoLaryngo, Ambu, Copenhagen, Denmark) attached to visualize the oropharyngeal swab technique on a video monitor (aView, Ambu, Copenhagen, Denmark) during the procedure (see Figure 1). The NPS was performed on a 3D-printed simulator based on real CT scans with correct anatomic landmarks [17]. The 3D simulator model was slightly modified for the project as we added ears on the simulator head, which is important guidance for the nasopharyngeal sample (see Figure 1). The URTS teaching material, checklists and the 3D printed model is made freely available on the following website: http://www.urt-sample.com (accessed on 25 October 2021).

### 2.4. OPS and NPS Training of the Group of Novices

All the novices received a total of 30-min training by a specialist in otolaryngology–head and neck surgery (AB). A power-point presentation demonstrated upper-airway anatomy and the steps in the techniques for performing OPS and NPS. The novices then observed a video from WHO demonstrating the technique on a real person [18]. The training session ended with hands-on practice of OPS and NPS on the simulation models described previously with feedback from the teacher. A post-training assessment was conducted with a theoretical MCQ test followed by an assessment of OPS and NPS performance on the simulators without any guidance from the teacher.

### 2.5. Competence Assessment

The novices answered the MCQ test and performed OPS and NPS on the simulators before (pretest) and again after training (posttest). The expert group also answered the MCQ and performed OPS and NPS on the same simulators, and all performances were video recorded for subsequent assessment with the URTS checklist. A nurse (RCE) and resident in otolaryngology—head and neck surgery (LH) rated all the videos independently. Prior to the rating, a 30-min rater training session was conducted. They watched four pilot videos (not being a part of the assessment videos used in the study) and discussed any disagreement in their URTS checklist ratings. Then, the two raters assessed all the videos using the URTS checklist in a randomized order and blinded to who had received training.

### 2.6. Statistics

To evaluate internal structure of the constructed MCQ, item statistics including item difficulty index and item discriminatory index were calculated. MCQs with appropriate testing properties were selected to be included in the test and internal consistency reliability was calculated as Cronbach’s alpha. Inter-rater reliability using intraclass correlation coefficients, absolute agreement definition, was used for measuring inter-rater reliability, and Cronbach’s alpha was used to evaluate the internal structure of the final items on checklists.

The “relations to other variables” validity evidence were explored by comparing the MCQ and checklist scores between groups. The test scores discriminatory abilities between experts and untrained/trained novices were explored with independent samples t-tests and the effect of the training was determined using a paired samples t-test.

The contrasting groups’ method [19] was used to explore the “test consequences” and establish the pass/fail levels of both tests to determine who is ready for independent practice. We examined the consequences of these criteria for both tests using the post-course test scores to determine the proportion of newly trained non-healthcare professionals having completed the combined theoretical and hands-on course that passed and failed the tests, respectively.

All the statistical analysis was performed using a statistical software package (PASW, version 27.0; SPSS Inc., Chicago, IL, USA), and two-sided significance levels of 0.05 were used for all analyses.

## 3. Results

Twenty-four novices without any healthcare education agreed to participate in the study and 16 experts who were registrars or consultants in otolaryngology—head and neck surgery working either at the Department of Otorhinolaryngology, Head and Neck Surgery at Zealand University Hospital, Køge or Rigshospitalet, Copenhagen, Denmark. All 24 novices were both pre- and post-tested while the 16 experts only were tested once. In total, results from 64 URTS MCQ tests and 128 videos of OPS and NPS performance were available for assessment with the URTS checklist.

### 3.1. Content Evidence

The URTS MCQ and the URTS checklist items were based on the WHO recommendations and revised based on content matter expert reviews. Based on the comments by the expert review, nine of the fourteen MCQ questions were revised, and one new item was added. Item analysis showed that 12 out of 15 items had acceptable item discriminatory indices of 0.18 or above, whereas the remaining three had indices below 0.04 and were removed from the final test. Five of the 17 URTS checklist items for OPS and NPS performance were revised to the final version approved by all the experts and used for the study (see Appendix A).

### 3.2. Response Process

All MCQ tests were administered in a standardized fashion without allowing the participants any guidance. After completing a half-hour rater training, the raters were able to conduct assessments using both OPS and NPS checklists without any missing values. The two raters reported no difficulty in performing ratings in the simulated context despite dissimilarities between the simulated and clinical procedure (e.g., no head movements during insertion of swabs or interaction with the patient).

### 3.3. Internal Structure

The internal consistency reliability of the MCQ test was 0.76. The URTS checklist score demonstrated good inter-rater reliability for OPS and NPS skills assessment of 0.86 and 0.87, respectively. Internal consistency reliabilities for both instruments were high, Cronbach’s alpha 0.92 and 0.93, respectively.

### 3.4. Relations to Other Variables

The URTS MCQ-test score was significant higher for the group of experts compared to the group of novices without training, *p* < 0.001 (see Table 1 and Figure 2). The novices scored higher after training (*p* < 0.001) but still significantly lower than the experts (*p* = 0.008).

The URTS checklist score was significantly higher for the group of experts compared to the group of novices without training, *p* < 0.001 for both OPS and NPS (see Table 1). The novices scored higher after both NPS and OPS training (*p* < 0.001) but not significantly lower than the experts when the OPS score (*p* = 0.14) and the NPS score (*p* = 0.49) were compared (see Table 1 and Figure 2).

### 3.5. Test Consequences

The contrasting groups’ method established an MCQ-test pass/fail score of 91% and a checklist pass/fail score of the 76% for OPS and 61% for NPS, see Figure 2. The consequences of the standard setting were that thirteen of the trained novices failed MCQ-test while only one failed the competency-based assessment with the URTS checklist (see Figure 2 and Table 1).

## 4. Discussion

This study provided validity evidence supporting the use of the developed URTS MCQ and checklist for the assessment of OPS and NPS skills. We found a high inter-test and inter-rater reliability, a significant discrimination between experts and novices’ URTS scores, and established pass/fail scores for URTS MCQ and checklists.

Sources of validity evidence were explored in this study following Messick’s validity framework. The content of the checklist was based on literature review and multi-speciality expert consensus. The response process was ensured by controlled testing environments, rater training, and the anatomical correct simulators used in the testing setup. A high inter-rater and inter-item reliability supported the internal structure validity evidence for both the MCQ-test and checklists. Relations to other variables were supported by the URTS MCQ and checklist ability to differentiate between novice and expert experience levels and the ability to measure an effect from 30 min training of the novices. The consequences of the test were explored by a standard setting establishing a meaningful pass/fail score. The finding that a group of trained novices did not obtain a significant lower URTS checklist score to compare to a group of experts may be surprising at first. However, previous studies have shown that experts often do not adhere strictly to detailed checklists compared with trained intermediate learners [20] although they are better at providing accurate overall diagnoses. However, COVID-testing is a relatively simple procedure requiring limited and relatively elementary knowledge and skills, which may indicate a short learning curve that combined with the standardized simulated model may have resulted in some level of ceiling effects. The good effect of a short simulation-based URTS training is also comparable to other studies with similar findings [21,22]. This indicates that URTS is a well-defined procedure where non-healthcare educated workers can perform URT samples comparably to experts if they receive focused and standardized training [23].

In this study, we assessed both OPS and NPS performance as WHO recommends combining both samples for each case to increase the diagnostic sensitivity [3]. In contrast, CDC primarily recommends a single NPS sample while other studies have questioned the OPS sensitivity [4,24,25]. If a single NPS specimen is used for testing purpose, the NPS checklist can just be used, and the OPS checklist can instead be omitted. The URTS assessment was developed in relation to the urgent need during the COVID-19 pandemic, but its use is not limited to coronavirus diagnostic but can also be used for other upper respiratory infectious diseases.

We found a high inter-rater reliability between the nurse and otolaryngologist checklist ratings, supporting the use of nurses instead of physicians for the practical testing and training of the large groups of workers involved in mass testing. Therefore, we believe the URTS MCQ-test and checklists can be used as part of training of new health-care workers at the outpatient COVID-19 testing sites and at the hospital. Further, it can also be used for training of laypersons needed in mass-testing strategies where the limitation of health workers during a pandemic will not be able to cover the test needs. The checklist was made in an OSCE format and could also be used as a part of clinical skills assessment at medical and nursing schools.

The findings of the study should be interpreted in accordance with the following limitations. The study was performed in a simulation-based setting, and we cannot know if the skills are directly transferable to how they would collect URT-specimens in an outpatient test site. However, a large body of existing literature supports the use of simulation-based assessment and training of clinical skills, and numerous studies have shown good correlations between performances in the simulated and clinical setting [26,27]. Moreover, in a clinical setting it will not be possible for the rater to see the intraoral OPS sampling technique which we instead could visualize with a video endoscope in the simulated setting for more complete assessment of sample technique (see Figure 1). A critical part of the NPS specimen collection is whether the swab reach the nasopharyngeal wall which require an insertion length between 8–11 cm [9]. However, the rater could not directly visualize if the swab reached the nasopharyngeal wall during the test participants performance on the NPS simulator model. Nevertheless, as the model has known realistic anatomy proportions it could be seen if the swab was inserted to high and trapped before at the midturbinates instead, see Figure 1 [9,17].

The aim of this URTS skills assessment is to ensure URT-specimens collection performance performed with correct infection control precautions. However, the URTS-skills assessment do not cover the full infection prevention and control competencies needed for health workers within infectious diseases and it should be covered by focused training and competency assessment within this area [28]. Further, the URTS-checklist focuses on the skills assessment needed for collecting specimens for SARS-CoV-2 molecular testing. Therefore, the NPS checklist should be combined with assessment of the on-site handling of the lateral flow immunoassay if it is used as a part of competency-based training in antigen testing [29].

The use of assessment instruments with established validity evidence is a cornerstone in the provision of competency-based URT-specimen collecting training of healthcare workers during the pandemic. However, until now the training and assessment of personnel performing URT-specimen collecting have received limited attention despite its importance for overall diagnostic accuracy of the tests used. A study found that NPS simulation training significantly can improve healthcare workers’ confidence, but the change performance was not assessed [30]. There has been a large emphasis on comparing the diagnostic accuracy of different test types (e.g., antigen vs. molecular tests) and different sampling sites (e.g., nasopharyngeal vs. oropharyngeal vs. saliva), and more focus should also be on the URT-specimen collecting technique and training [31,32]. Future research and practice must incorporate descriptions of how the sampling procedure is performed and the test personnel are assessed and trained to ensure standardization and reproducibility of test design. This also applies to self-collected URT samples which can be an effective method for mass testing in future [33,34].

## 5. Conclusions

We established validity evidence supporting the use of a theoretical and practical test for upper respiratory specimens’ collection skills based on the recommendations by the WHO. Further, the results indicate that non-healthcare educated workers can perform URT samples comparably to experts after a short and focused simulation-based training session. This is the first step to enable competency-based training of the workers in the COVID-19 test centers as well as for health-care workers performing URT samples for other infectious diseases.

## Figures and Tables

**Figure 1 diagnostics-11-01987-f001:**
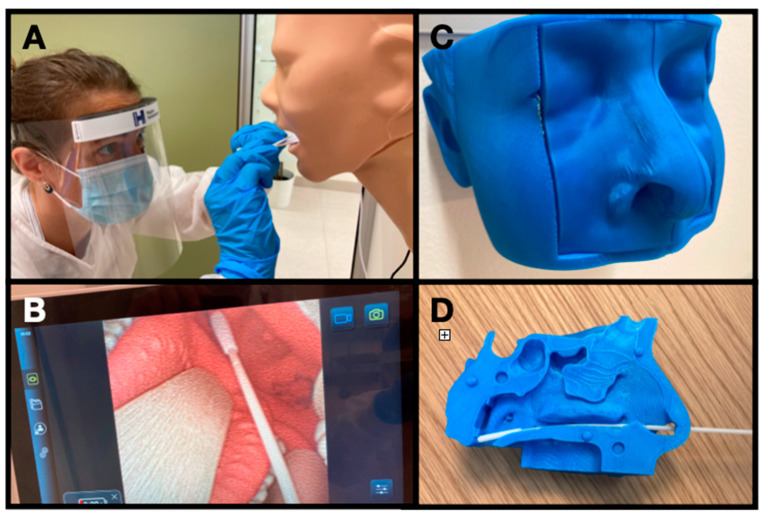
Test setup in a simulation setting. (**A**) Oropharyngeal sample on a life-sized airway demonstration model. (**B**). Videoendoscopic recordings of oropharyngeal sample technique. (**C**) Three dimensional printed model for nasopharyngeal sampling. (**D**). Inside the 3D-printed model with correct anatomic landmarks based on CT scans.

**Figure 2 diagnostics-11-01987-f002:**
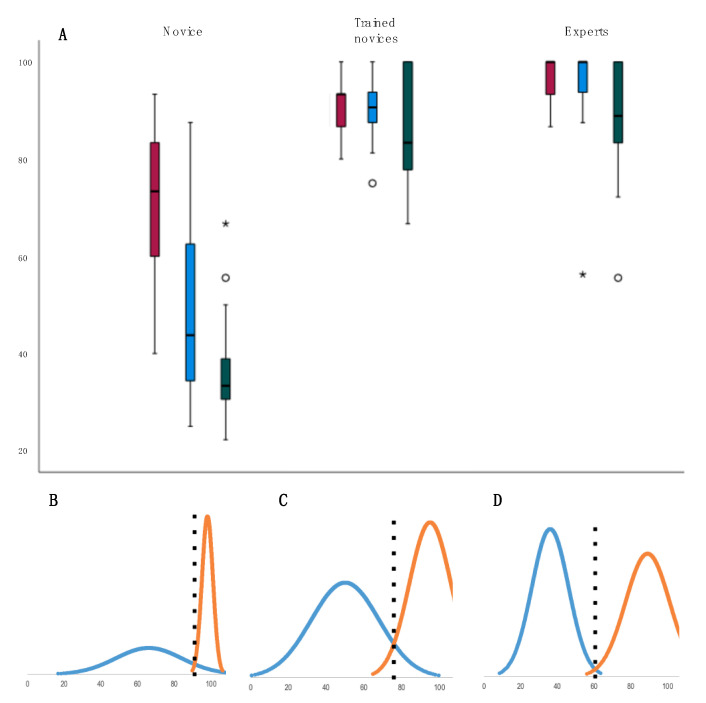
Boxplots and standard setting. (**A**). Boxplot with the mean MCQ (red), OPS checklist (blue) and NPS checklist (green) URTS scores at the Y-axis (percent of total number of possible correct answers) for the different groups. The central bar in the box represents the median, the box represents the interquartile range, and the whiskers represent the range. Outliers are plotted as individual points (*). (**B**–**D**). Standard setting with the contrasting groups’ method and normally distributed curves representing the score of the novice (blue), and expert (yellow) groups. The black dotted vertical line represents the pass/fail cut-off score for the MCQ test (**B**), OPS checklist score (**C**) and NPS score (**D**).

**Table 1 diagnostics-11-01987-t001:** The URTS MCQ-test and checklist scores.

	Novice Group (*n* = 24)	Group of Trained Novices (*n* = 24)	Expert Group (*n* = 16)	Pass-Fail Score
MCQ score%, mean (SD)	66 (18)	93 (8)	98 (3)	91
OPS-checklist, Mean (SD)	50 (18)	91 (6.9)	95 (11)	76
NPS-checklist, Mean (SD)	36 (10)	86 (11)	89 (12)	61

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
