# Peer review of "Valid and Reliable Assessment of Upper Respiratory Tract Specimen Collection Skills during the COVID-19 Pandemic"

_diagnostics, 2021, doi:10.3390/diagnostics11111987_

Round 1
Reviewer 1 Report
This is generally a well-written and comprehensive article regarding the validity of theoretical and practical skills tests for obtaining nasopharyngeal and oropharyngeal specimen collection for COVID-19 testing. The results are clearly and transparently presented and the set goals correspond to the conclusions. I consider that the findings are interesting and that the results obtained can make significant contributions to further large studies. However, I suggest better outline the subchapter related to the limitations of the study. Also, I recommend better detail the results of their study compared to other research conducted on this topic, in order to highlight the original elements of the study.
Author Response
Thank you for your very nice comments. Please see answers to your questions in the red attached.

Reviewer 2 Report
Todsen and colleagues report the establishment and validation of a training assessment to non-professionals on upper respiratory tract sampling. They found that novices performed comparably to experts after a 30-minute training with slides and videos, as measured in simulated settings. The training material can be used to training more non-healthcare workers to meet to high demand of workers to conduct high-quality sampling work in COVID-19 diagnostics. The work is well-designed and the manuscript is clearly written. A few comments to the authors to address -
- The training materials, including slides and MCQs, should be made available.
- Describe in more details how the numbers of 64 MCQ tests and 128 videos of oropharyngeal and nasopharyngeal swab sampling performance were derived, given test sizes of 24 novices and 16 experts in the study.
- Checklist: (1) Are there any quantitative criteria in providing a general assessment score from “Bad” to “Excellent”? (2) It appears some items, such as #1 on correct use of PPE, are less influential to the quality of sampling. Was each item in the checklist weighted differently? If no, why not?
- Based on the description provided in the section “3.1 Content evidence”, the finalised checklist should have contained 17 items, but only 8 were listed in Appendix A. Please clarify.
- Did the authors gauge the possible relationship of education level of novices and their performance after training? In particular, it would be of interest to know whether those with higher education background (e.g., holders of non-healthcare—related degree) had superior performance than others after the training.
- Figure 2: Y-axis labels are missing.
- Table 1: This table seems just a repeat of Figure 2. It is not clear about the added value of this table, except that it shows means rather than medians of scores. Indicate test size of each subgroup in Figure 2/Table 1.
Author Response
Thank you for your review. Please see the answers in red attached.
